# Discovering the Triad between Nav1.5, Breast Cancer, and the Immune System: A Fundamental Review and Future Perspectives

**DOI:** 10.3390/biom12020310

**Published:** 2022-02-15

**Authors:** Harishini Rajaratinam, Noor Fatmawati Mokhtar, Nurul Asma-Abdullah, Wan Ezumi Mohd Fuad

**Affiliations:** 1School of Health Sciences, Health Campus, Universiti Sains Malaysia (USM), Kubang Kerian 16150, Kelantan, Malaysia; harishinir@student.usm.my (H.R.); nurulasma@usm.my (N.A.-A.); 2Institute for Research in Molecular Medicine (INFORMM), Health Campus, Universiti Sains Malaysia (USM), Kubang Kerian 16150, Kelantan, Malaysia; fatmawati@usm.my

**Keywords:** *SCN5A*, Nav1.5, neonatal Nav1.5, immune system, breast cancer, immunotherapy

## Abstract

Nav1.5 is one of the nine voltage-gated sodium channel-alpha subunit (VGSC-α) family members. The Nav1.5 channel typically carries an inward sodium ion current that depolarises the membrane potential during the upstroke of the cardiac action potential. The neonatal isoform of Nav1.5, nNav1.5, is produced via VGSC-α alternative splicing. nNav1.5 is known to potentiate breast cancer metastasis. Despite their well-known biological functions, the immunological perspectives of these channels are poorly explored. The current review has attempted to summarise the triad between Nav1.5 (nNav1.5), breast cancer, and the immune system. To date, there is no such review available that encompasses these three components as most reviews focus on the molecular and pharmacological prospects of Nav1.5. This review is divided into three major subsections: (1) the review highlights the roles of Nav1.5 and nNav1.5 in potentiating the progression of breast cancer, (2) focuses on the general connection between breast cancer and the immune system, and finally (3) the review emphasises the involvements of Nav1.5 and nNav1.5 in the functionality of the immune system and the immunogenicity. Compared to the other subsections, section three is pretty unexploited; it would be interesting to study this subsection as it completes the triad.

## 1. Introduction

Nav1.5 (encoded by the *SCN5A* gene) is one of the nine members of the VGSC family. VGSCs are heteromeric membrane protein complexes. The VGSC structure is composed of one pore-forming α subunit and smaller β subunits. In total, there are nine α subunits (Nav1.1–Nav1.9) and four β subunits (β1–β4) [1] (Figure 1). In standard settings, VGSCs are responsible for the inward sodium current (I_Na_) in excitable cells. As such, they induce fast depolarisation, thereby initiating a potential for the cells to react [2,3].

In terms of conformational change, VGSC transits between three distinct conformational states whenever there is a change in the membrane potential (depolarisation). These unique conformational figures are known as resting (closed), activated (open), and inactivated (closed) states [5]. Refractory is a term often associated with the functionality of VGSC. The term ‘refractory’ refers to a period of recovery from inactivation during which the channel is unable to open in response to depolarisation [6,7].

The Nav1.5 channel typically carries an inward sodium ion current (I_Na_) which determines the sodium ion influx that functions to depolarise the membrane potential during the upstroke of the cardiac action potential [8]. Nav1.5 is important in regulating normal cardiac development, maintaining the heart‘s rhythm, and preventing various cardiac-related diseases [3]. The Nav1.5 channel is encoded by the *SCN5A* gene [9,10]. 

Alternative mRNA splicing allows further functional variation among α subunits of the VGSC family [11]. The neonatal isoform of Nav1.5, or neonatal Nav1.5 (nNav1.5), is produced as a result of VGSCα alternative splicing at domain 1: segment 3 (D1:S3) [12]. In breast cancer, the up-regulation of such an alternative splice-variant portrays onco-foetal gene expression since nNav1.5 would normally be expressed only during the foetal stage of human development [13,14]. The molecular differences between these two isoforms include the position of Nav1.5 (3′) and nNav1.5 (5′) on exon 6 and the seven amino acid changes in the sequence of nNav1.5 protein compared to Nav1.5 (Figure 2). The location of the alternative splicing is in the S3 and S4 regions of D1 in the protein, including the extracellular S3–S4 linker (Figure 2). The sequence differences between the two isoforms are located mainly in the C-terminal end of D1:S3 and the S3/S4 linker region of the channel, close to the four positively charged residues of the voltage-sensing S4 [12], as depicted in Figure 2.

In short, the alternative splicing replaces a conserved negative aspartate residue in the ‘adult’ isoform with a positive lysine [12]. Due to the electrophysiological changes contributed from the Nav1.5 D1:S3 splicing, the charge reversal in nNav1.5 modifies the kinetics of the channel which results in the prolonged resultant current and thus causes an increased intracellular sodium ion (Na^+^) influx [12].

Aside from charge reversal and amino acid substitutions, there is one other feature that distinguishes Nav1.5 from nNav1.5. This feature is known as the ability to resist the suppressive effects of acidification. According to Onkal et al. [15], nNav1.5 showed more resistance against the suppressive effects of acidification than Nav1.5. Such resistance was associated with the difference in the charged amino acids of nNav1.5 and Nav1.5, which was mentioned earlier. It was postulated that the replacement of negatively charged aspartate to the positive lysine in the DI: S3–S4 region may decrease the effects of protonation on the activation of nNav1.5 [15]. Additionally, the residuals of acidic influence in S3–S4 may serve as crucial determinants of cationic effects pertaining to channel gating [16,17].

Nav1.5 and nNav1.5 have exhibited functional roles in assisting cancer progression, especially in breast cancer. The detection of these sodium channels, in association with breast cancer, was conducted using various modes of experiments such as in vitro [18], in vivo [19], and via the use of clinical samples [13,20]. In the midst of these oncological-based studies, another branch of Nav1.5 and nNav1.5 began to expand. This branch focuses on the involvement of Nav1.5 and nNav1.5 in the immune system’s functionality. Since there is a proven connection between breast cancer and the immune system [21], it would be interesting to review and discuss evidence supporting the connection between Nav1.5 (nNav1.5), breast cancer, and the immune system. 

Therefore, in the current review, we have proposed and reviewed the triad encompassing Nav1.5, breast cancer, and the immune system (as depicted in the graphical abstract). To the best of our knowledge, there is no such review published to address the unique triad we propose in the present article. The available review papers only focus on the metastatic capacity of Nav1.5 and nNav1.5 in potentiating breast cancer metastasis [22] and the pharmacological aspects of these sodium channels [23]. 

To obtain a clearer picture of the triad, we have subdivided the review into three major sections, which are (1) the roles of Nav1.5 and nNav1.5 in potentiating the progression of breast cancer metastasis; (2) the general connection between breast cancer and the immune system; and (3) the involvements of Nav1.5 and nNav1.5 in the functionality of the immune system and the immunogenicity of nNav1.5. Compared to the other subsections, section three is underexploited, and it would be interesting to study this particular subsection as it completes the triad between Nav1.5, breast cancer, and the immune system. 

Finally, we aimed to reassemble the triad and highlight prospects that could serve as future perspectives to aid breast cancer immunotherapy. 

## 2. The Roles of Nav1.5 and nNav1.5 in Potentiating the Progression of Breast Cancer

### 2.1. Introduction to Breast Cancer 

Over the years, breast cancer has been one of the most diagnosed forms of cancer that occurs among females worldwide. According to the latest GLOBOCAN report released by the International Agency on Cancer Research (IARC) in 2020 [24], breast cancer has the highest percentage of incidence (24.5%), followed by colorectal cancer (9.4%), lung cancer (8.4%), and cervical cancer (6.5%). The report added that breast cancer holds the record for the highest number of prevalent cases in 5 years (30.3%) and the number of death cases (15.5%). 

Breast cancer subtypes can be classified based on histological, molecular, and functional classifications [25]. In terms of histological classification, breast cancer cases can be classified as invasive ductal carcinoma (IDC) and invasive lobular carcinoma (ILC) [26]. Under histopathological conduct, several characteristics are taken into consideration to determine whether the tumour is either more likely to be IDC or ILC. These characteristics include cell type, number of cells, type of secretion, location of secretion, immunohistochemical analysis, and architectural features [26].

Besides histological classification, another option is molecular classification which is conducted based on the intrinsic molecular subtypes of breast cancer and could be identified using gene profiling (microarray) [27,28]. Initially, only five distinct subtypes of breast cancer were identified: Luminal A, Luminal B, HER2 enriched, basal-like, and normal breast-like. In general, tumours that neither express oestrogen receptor (ER), progesterone receptor (PR), nor human epidermal growth factor receptor 2 (HER2) are defined as triple-negative breast cancer (TNBC) [29]. TNBC subtypes are subject to poor prognosis as conventional treatments such as hormonal therapy and HER2 targeted therapy have to be ruled out due to the absence of the corresponding receptors. Similar to the basal-like subtype, TNBC exhibits an aggressive nature compared to the other subtypes. In 2008, Bertucci et al. [30] highlighted that TNBC and basal-like breast cancer are not the same entity. It is important to address that TNBC does not form a homogeneous group when analysed by gene expression profiling, and in contrast, the basal-like subtype forms a homogeneous group of tumours with a similar gene expression profile related to prognosis and therapy response [30,31].

In 2011, the TNBC subtype was further differentiated. Six different TNBC subtypes were identified, which were basal-like 1 (BL1), basal-like 2 (BL2), mesenchymal (M), mesenchymal stem-like (MSL), immunomodulatory (IM), and luminal androgen receptor (LAR) [32]. An alternative classification divides TNBC into BL1 and BL2, M, and LAR [33]. Recently, Wang et al. [34] reported that BL1, BL2, and IM, can be further stratified on the basis of intrinsic oncogenic alterations. Even though TNBC is not a homogeneous breast cancer disease entity, a substantial fraction of this subtype belongs to the basal-like tumour type (which form a homogeneous group). Thus, the overall poor prognosis of TNBC may be a result of this basal-like subgroup and triple negativity may be seen more as a symptom than as a separate entity of breast cancer [35]. 

### 2.2. The Role of Nav1.5 and nNav1.5 in Breast Cancer Metastasis

For almost a decade, the link between Nav1.5 (and nNav1.5) and breast cancer metastasis has been substantially studied. As a result, it was revealed that both Nav1.5 and nNav1.5 support the growth of breast carcinoma, especially the aggressive and metastatic subtypes. There are remarkable in vitro and in vivo pieces of evidence that suggest the functional upregulation of these proteins in breast cancer may lead to its progression.

An early study by Roger et al. [36] revealed the presence of a fast-inward sodium current (I_Na_) in highly metastatic breast cancer cells, MDA-MB-231. The finding was among the earliest studies that linked sodium channels with the invasion capacity of breast cancer cells. The role of nNav1.5 in promoting migration in the MDA-MB-231 cells was demonstrated by Isbilen et al. [37], whereby the introduction of docosahexaenoic acid (DHA) or omega-3 reduced migration in the cells via the downregulation of nNav1.5 expression. DHA (22:6n-3) is a long-chain polyunsaturated fatty acid. Isbilen et al. [37] demonstrated that DHA or omega 3-induced suppression of cellular migration might occur via the downregulation of nNav1.5 mRNA expression and the nNav1.5 functional protein. The localisation of VGSC channels in the lipid rafts [38] may be affected by the actions of omega-3 [39]. A study by Blanckaert et al. [40] demonstrated that the proteins from the crude membrane preparations of MDA-MB-231 cells, when treated with DHA, showed an increase in the expression of type Ⅱ cytoskeletal 1 (KRT1). KRT1 was previously shown to interact with tyrosine kinase Src via binding to integrin β1 [41]. In the context of breast cancer, the KRT1 level is strongly reduced in breast cancer cells which assists them in achieving a metastatic phenotype [42]. Interestingly, an indirect association between KRT1 and Src was reported [41], as well as the relationship between Src and Nav1.5 by Andavan et al. [43] and Brisson et al. [9]. Therefore, we hypothesise that the downregulation of nNav1.5 might occur via the indirect effects of DHA on Src through KRT1. To the best of our knowledge, there is no direct evidence pertaining to the mechanism on how DHA or omega-3 downregulates nNav1.5. 

In the following year, a comparison study was conducted by Brackenbury et al. [18] whereby it was revealed that there was a higher level of nNav1.5 expression in migrating MDA-MB-231 breast cancer cells compared to non-migrating breast cancer cells, MCF-7. The finding was further validated by the introduction of nNav1.5 siRNA (small interfering RNA) that contributed to the reduction of migration in MDA-MB-231 cells [18].

In a study by Erdogan and Ozpolat [44], MDA-MB-231, MCF10A, and MCF-7 were utilised to demonstrate the upregulation of Nav1.5 and nNav1.5 in promoting breast cancer metastasis. It is vital to note that these cells have different characteristics and metastatic potential, as well as a unique profile of hormone receptors (or immunohistochemical profile) [45]. In this study, siRNAs against both Nav1.5 and nNav1.5 were introduced to the cells in the form of treatments. It was found that MDA-MB-231 cells exhibited higher mRNA expressions of Nav1.5 and nNav1.5 as compared to MCF-7 cells. Additionally, the introduction of Nav1.5 siRNA led to a significant reduction in the proliferation, invasion, migration, drug resistance, and capacity to form colonies. Targeting Nav1.5 and nNav1.5 by systemically addressing the use of siRNA is a fascinating and convenient approach. Interestingly, Nav1.5 siRNA did not exert any effects that may affect cell proliferation in the normal breast epithelium cell line, MCF10A [44]. Correspondingly, Fraser et al. [20] found that no inward current was observed in MCF10A. 

Most of the discussed in vitro studies indicate that aggressive breast carcinoma cell lines such as MDA-MB-231 are found to possess a higher expression of Nav1.5 and nNav1.5 as compared to weakly aggressive cell lines such as MCF-7. The presence of Nav1.5 and nNav1.5 in these aggressive cell lines are associated positively with various metastatic traits such as migration and invasion [1,20,46,47,48,49]. 

The simplicity of an in vitro assay has opened the path for various pharmacological tests as well. Using in vitro assays, researchers have investigated the effects of VGSC blockers on the migration and invasion capacity of metastatic breast cancer cells. As in the case of Fraser et al. [50], caffeic acid phenyl ester was introduced as a potential VGSC blocker that reduced invasion in MDA-MB-231 cells. Phenytoin has also been reported to potentially downregulate the expression of Nav1.5 in highly metastatic breast cancer cells using in vitro assay [47,48]. In addition to caffeic acid phenyl ester and phenytoin, ranolazine also showed an inhibitory effect on Nav1.5 mediated current and reduced metastasis in in vivo models [51]. 

However, these in vitro studies cannot be assumed to ultimately display Nav1.5 and nNav1.5 as potential targets to combat breast cancer since the findings are quite fundamental. The animal models (in vivo) and human models are far more complex than a single cell line since various body systems are involved and may influence breast cancer progression.

In a breakthrough study by Nelson et al. [19], the metastatic capacity of Nav1.5 was highlighted extensively using an animal model. The study found that the downregulation of Nav1.5 expression by using lentiviral shRNA significantly reduced tumour growth, local invasion, and tumour metastasis in the orthotopic breast cancer mice model. Additionally, Nelson et al. [19] also revealed the potential of phenytoin as a pharmacological drug that can be used to reduce breast cancer metastasis in in vivo models. The study implies that Nav1.5 possesses the ability to promote metastasis to the surrounding distant organs such as the liver, lungs, and spleen despite the presence of complex body systems in animal models [19]. 

In terms of clinical studies, a very limited number of studies have been conducted to investigate the expression of Nav1.5 and nNav1.5 in human breast tumour biopsy samples. In a study by Fraser et al. [20], it was found that the expression of nNav1.5 was markedly upregulated in human breast cancer biopsy sections compared to normal breast tissue. This evidence suggests that the high expression of nNav1.5 occurs not only in human breast cancer cells (in vitro) but also in primary breast cancer tissues (clinical). Furthermore, the expression of nNav1.5 measured in these breast cancer tissues was found to be strongly related to lymph node metastasis (LNM). Interestingly, there were no cases of nNav1.5 negative/LNM positive reported, implying that metastasis to lymph nodes never occurred when nNav1.5 was not expressed in the primary tumour [20]. The promising finding that links the expression of nNav1.5 with lymph node metastasis may suggest that VGSCs could act as an independent prognostic variable in breast cancer progression (metastasis). 

A more extensive study was conducted by Yamaci et al. [13], as compared to the one by Fraser et al. [20], in which the expression of nNav1.5 in human breast cancer tissues was investigated on a larger scale and using a bigger sample size. Yamaci and colleagues [13] compared the expression of nNav1.5 between breast cancer biopsies and normal human adult tissues. The study reported that there was a higher expression of nNav1.5 in the epithelial cells of breast cancer tissues compared to those obtained from a normal breast biopsy. Surprisingly, some normal breast ductal epithelial cells also portrayed vague nNav1.5 immunoreactivity [13].

In relation to the estrogen receptor (ER) status, Yamaci et al. [13] found that in all cases of ER-alpha negative (ERα-), there was a positive presence of nNav1.5 protein. In contrast, nNav1.5 was absent in all cases of ER-alpha positive (ERα+). The study then shifted its focus onto the subcellular localisation of the nNav1.5 protein (in plasma membrane and/or cytoplasm). It was found that, in the cases of ERα-, the protein expression of nNav1.5 was restricted to the plasma membranes whereas in cases of ERα+, the protein expression of nNav1.5 was observed either in the plasma membrane and/or cytoplasm, and it was not restricted to the plasma membrane. Interestingly, the study noted that there was a significant positive correlation between the protein expression of nNav1.5 and the absence of ERα [13]. Currently, there is no specific evidence that directly addresses the exact functional locations of Nav1.5 and nNav1.5 in the cytoplasm of ERα+ cases. However, evidence addresses that the ion channels may be bound to the organelle within the cytoplasm [52]. Generally, the life cycle of Nav1.5 starts with the transcription of the *SCN5A* gene and RNA processing. Then, the Nav1.5 proteins are synthesised at the rough endoplasmic reticulum and follow the secretory pathway in order to reach the plasma membrane where the proteins exert their respective function [53]. 

The exclusive existence of nNav1.5 in the plasma membrane, in the absence of ERα, supports the notion that the nNav1.5 channel protein may possess a functional role in promoting breast cancer metastasis which resonates with the earlier findings from the in vitro studies that have been conducted using MDA-MB-231.

Note that the MDA-MB-231 human breast cancer cell line has been vastly used to thoroughly study the relationship between Nav1.5/nNav1.5 and breast cancer metastatic capacity. This highly metastatic cell line has been portrayed to possess a high expression of nNav1.5 and does not have any hormone receptors (such as ER and PR) and HER2 (otherwise known as triple-negative). According to Mokhtar et al. [54], the expression of nNav1.5 was upregulated in both MDA-MB-231 cells and the 4T1, murine mammary cancer, cell line. Similar to MDA-MB-231, the 4T1 cell line is also referred to as a triple-negative cell line. The 4T1 cells share substantial molecular features which are similar to those observed in human TNBC [55]. Putting all these facts into one context, we may conclude that the functional nNav1.5 expression is upregulated in metastatic breast cancer cells that do not possess hormone receptors, especially ER. On the contrary, weakly metastatic breast cancer cells do not possess any functional VGSC due to the presence of ER, which may explain why the expression of Nav1.5 and nNav1.5 are higher in MDA-MB-231 as compared to MCF-7. Some initial discoveries and postulations suggest that, since ion transports are generally controlled by hormones and growth factors which are also known as key participants in mainstream cancer progression mechanisms [56,57], there is an association between the presence of ion transports and the progression of metastasis in cancer. The concept of the inverse relationship between genomic steroid hormone sensitivity and functional VGSC expression was highlighted by Fraser et al. [57].

### 2.3. Models Proposed as Mechanisms to Explain How Nav1.5 Promotes the Metastatic Behaviour in Aggressive Breast Cancer Cells

There are several models proposed to explain how VGSCs, particularly Nav1.5, promote metastatic behaviours in breast cancer. In breast cancer metastasis, the presence of invadopodia is a key determining factor in ensuring the degradation of the extracellular matrix and the invasion/migration of cancer cells. Invadopodia are actin-rich organelles that present as protrusions in the plasma membrane [58]. Invadopodia adopt a pseudotubular structure in order to surround and digest the extracellular matrix (ECM). The plasma membrane of the invadopodia is rich in Nav1.5 and sodium/hydrogen exchanger 1 (NHE-1) channels. These channels are present in close proximity to the dimers of caveolin-1 protein in the areas of lipid rafts. Caveolin-1, NHE-1, and Nav1.5 coprecipitate when they are treated with antigens [59].

According to Brisson et al. [9], Nav1.5 interacts with NHE-1 in the focal ECM degradation sites, which correspond to invadopodia that contain the caveolin-1 protein. Nav1.5 activity controls src kinase activity which can cause the phosphorylation of the actin nucleation-promoting factor known as cortactin. Thus, the situation is able to initiate the polymerisation of actin filaments [9] and the allosteric modulation of NHE-1, consequently enhancing the invadopodial proteolytic activity that contributes to breast cancer invasion (Figure 3 and Figure 4). The activation of proteolytic enzymes occurs via the extrusion of hydrogen ions or protons (H^+^) by NHE-1 into the extracellular space (Figure 3). The extrusion of protons lowers the pH level of the extracellular environment and the acidic environment favours the degradation of the ECM (Figure 4).

In an extension to the study performed by Brisson et al. [9], the CD44-src-cortactin signalling axis was investigated in conjunction with Nav1.5 (Figure 3). It is well established that the adherence of CD44 to its ligand leads to the activation of src and the phosphorylation of cortactin, as mentioned earlier [60]. CD44 has been known to increase the efficiency of distant metastasis in breast cancer [61]. Interestingly, Nelson et al. [19] found some substantial evidence linking these three components together, which are CD44, Nav1.5, and breast cancer. According to the study, the downregulation of Nav1.5 resulted in a decline in the expression of CD44, followed by a change in the cell morphology. The study implies that Nav1.5 (VGSC) may regulate invasion in breast cancer via the CD44-src-cortactin signalling pathway [19]. 

In the next model, we will look at how Nav1.5 modulates the pH level of the surroundings that contribute to the degradation of the ECM. The degradation of ECM is crucial for the invasion of cancerous cells. Since Nav1.5 and NHE-1 are co-expressed within the lipid rafts of the plasma membrane in cancer cells, the communication between Nav1.5 and NHE-1 plays an important role in regulating the pH of the surroundings. The sodium ion (Na^+^) influx contributes to an intracellular alkalisation and extracellular acidification adjacent to the plasma membrane. The Na^+^ influx through Nav1.5 promotes an increase of H^+^ efflux through NHE-1 (Figure 3). The imbalance in the number of Na^+^ ions and H^+^ ions give rise to an acidic pH value in the extracellular surrounding. Low pH contributes to the proteolytic activity of cysteine cathepsins that ultimately causes the degradation of the ECM and favours cancer cell invasion [9,46,62]. 

In an attempt to connect the dots, we noticed a pattern in these findings which associates NHE-1, CD44, and Nav1.5 together (Figure 3). According to a study by Chang et al. [63], there was a positive correlation between CD44 and NHE-1 in breast cancer cells. The downregulation of CD44 resulted in a drop in the expression of NHE-1 and inhibited metastasis in the MDA-MB-231 cell line. The study also added that the downregulation of CD44 prevented lung metastasis and the progression of breast tumours. Whilst in the study by Nelson et al. [19], the downregulation of Nav1.5 significantly decreased the breast tumour growth, local invasion, and metastasis to the lungs, liver, and spleen. Therefore, these findings suggest a modulation between Nav1.5, NHE-1, and CD44 which favours the progression of breast cancer. Chang et al. [63], added that the upregulation of CD44 and NHE-1 activity contribute to the enhanced expression of matrix metalloproteinases (MMPs) via the phosphorylation of ERK1/2 (Figure 3). 

Aside from that, Chioni et al. [64] added that the Na+ current conducted by Nav1.5 activates the phosphorylation of protein kinase A (PKA), which increases the expression of Nav1.5 at the transcriptional level (mRNA), but does not cause an increase in the total protein level. Such occurrence manipulates the distribution of Nav1.5 in the plasma membrane and cytoplasm of metastatic breast cancer cells such as in MDA-MB-231. The externalisation of Nav1.5 mostly takes place in the plasma membrane, leaving a reduced expression in the cytoplasm [64]. The externalisation of Nav1.5 in the plasma membrane ultimately leads to the potentiation of invasion and migration of breast cancer cells [64].

In addition to NHE-1, salt-inducible kinase-1 (SIK-1) also plays a prominent role in regulating Nav1.5-dependent epithelial-mesenchymal transition (EMT) and invasiveness in highly aggressive human breast cancer cell lines such as MDA-MB-231 (Figure 3). According to Gradek et al. [65], the retardation in the expression of SIK-1 potentiated the expression of Nav1.5 and the expression of EMT-promoting transcription factor SNAI1. The phenotypic transition of epithelial to mesenchymal morphology of cancerous cells is considered a critical event that supports the progression of breast cancer [66]. EMT is often navigated by various factors which include the Rho family of GTPases, particularly, Rac1 and Rac1b [67]. Evidence has shown that the Rac1 signalling pathway is involved in the regulation of EMT and cancer cell motility [68,69]. In relevance to our present review, Yang et al. [68] discovered that the activation of Rac1 by the Nav1.5 channel stimulates the acquisition of a motile phenotype and cell migration in breast cancer cells. The study also identified a mechanism where the activation of Rac1 manipulates cell migration in response to ionic and/or electrical changes in the microenvironment [70]. Nav1.5 dependent plasma membrane potential (Vm) depolarisation may control Rac1 activation via its interaction with phosphatidylserine [22] (Figure 3). 

In addition to Rac1, another member of the GTPase family, known as RhoA, has also been studied in association with Nav1.5. In general, it was suggested that the increase in the expression of RhoA contributes to cell migration leading to metastasis [71]. A study was conducted by Dulong et al. [72] where the position of RhoA in the regulation of expression and functional role of the Nav1.5 was investigated using MDA-MB-231 and MCF-7. The study revealed that the silencing of RhoA resulted in a reduction of the sodium current mediated by the sodium channel [72]. The silencing of RhoA also downregulated the expression of Nav1.5 at a transcriptional level. These two findings imply that RhoA applies a stimulatory effect on the production of an active form of Nav1.5 channel in the cancerous cell [72]. However, the silencing of Nav1.5 showed no effect on RhoA at a transcriptional level. Interestingly, the silencing of Nav1.5 resulted in a reduction of RhoA at a protein level in MDA-MB-231 cells [72]. From our understanding, we believe that the inhibition of Nav1.5 at a transcriptional level by siRNA targeting RhoA results in a functional knockdown of Nav1.5 since it disrupts the synthesis of the Nav1.5 protein. Such occurrence reflects an interdependency rather than a sequential activation pathway. The decrease in RhoA protein by siRNA against Nav1.5 reflects the positive feedback of Nav1.5 on RhoA which suggests that Nav1.5 might influence the stability of RhoA protein [72]. 

The next model proposed is the regulation of gene expression. In a study by House et al. [73], *SCN5A* emerges as a key regulator of the invasion gene network in colon cancer. This supports the notion that Nav1.5 may function as an early entry point in the signalling cascade that regulates invasion. The possible downstream gene network or ontology regulated by Nav1.5 includes the WNT signalling, the regulation of membrane and protease genes, calcium signalling, MAP kinase signalling and membrane remodelling, migration of cells, ectoderm development, response to biotic stimulus, steroid metabolism, and regulation of the cell cycle [1,73]. Even though this particular study does not directly reflect the gene expression pioneered by *SCN5A* in breast cancer, it provides an insight into the potential of *SCN5A* (Nav1.5) in promoting the invasion of cancerous cells by regulating crucial gene networks. 

The final model represents the association between VGSCs and breast cancer metastasis via angiogenesis. Angiogenesis can be defined as the development of new blood vessels that play a central role in providing nutrients and oxygen for the proliferating cancerous cells and maintaining the growth of carcinoma [74]. Sustained angiogenesis has also been recognised as one of the hallmarks of cancer. One study has shown that VGSCs can indirectly stimulate breast cancer metastasis by promoting angiogenesis or vascularisation. VGSCs are capable of upregulating the vascular endothelial growth factor (VEGF) signalling in endothelial cells that are present within the tumour environment [75]. The crosstalk between endothelial cells and malignant breast cancer cells within the tumour microenvironment is found to contribute to the development of neovascularisation which promotes metastasis. A study by Buchanan et al. [76] demonstrated that the angiogenic activity of malignant mammary epithelial cells is significantly enhanced by the presence of endothelial cells. The study found that when MDA-MB-231 cells are co-cultured with endothelial cells, the expression levels of VEGF and angiogenesis-promoting protein genes 2 (ANG2) were significantly increased. Recently, a study by Rajaratinam et al. [77] found that there was a significant inverse correlation between the expression of VEGF and anti-nNav1.5 antibodies in the serum of breast cancer patients who underwent treatment. This implies that during breast cancer therapy, VEGF expression was downregulated due to the positive impact of breast cancer therapy. This supports the indication that the neonatal isoform has the capacity to initiate VEGF-induced endothelial angiogenesis as the increased level of anti-nNav1.5 antibodies was parallel to the decrease in the level of VEGF [77].

An important study by Andrikopoulos et al. [75] revealed that different aspects of angiogenesis are controlled by different VGSC isoforms (different α subunits). As in the case of Nav1.5, it is responsible for endothelial cell proliferation and tubular differentiation. The introduction of Nav1.5 RNAi has led to the attenuation of phospho-ERK1/2 levels in response to VEGF. Hence, this proves that VGSCs (Nav1.5) potentiate angiogenesis via the VEGF-induced ERK1/2 activation. The classical PKCα isoform is a key regulator of the VEGF-induced ERK1/2 activation, which is crucial for endothelial proliferation and angiogenesis. It was further added by Andrikopoulos et al. [75] that VGSC may indirectly influence the translocation of PKCα to induce the ERK1/2 activation via the regulation of calcium ions. In contrast with cancer cells and macrophages, the accumulation of calcium ions takes place intracellularly in endothelial cells via the sodium-calcium exchanger (NCX). The exchange of sodium and calcium ions in the reverse mode of operation is reported to influence the activation of PKCα, thus potentiating the proliferation of endothelial cells to form blood vessels as demonstrated by Andrikopoulos et al. [75] using HUVEC cells (human endothelial cells). 

To the best of our knowledge, the role of Na+ ions from Nav1.5 as the driving force for the reverse mode NCX function is not yet fully understood. However, we have identified several ideas that may contribute to that point of view. As mentioned earlier, Andrikopoulos et al. [75] reported that VGSCs (particularly Nav1.5) are found to potentiate VEGF-induced ERK1/2 activation via the PKCα-B-RAF signalling axis. It was further deduced that such potentiation might be associated with VEGF-induced HUVEC depolarisation and intracellular calcium ions [75]. Further investigation by Andrikopoulos et al. [78] demonstrated that calcium ion influx via the reverse mode NCX acts as a determining factor in endothelial angiogenesis and barrier function in relation to thrombin. Recently, it was reported that nNav1.5 and VEGF-A are associated which each other in the context of colorectal cancer [79]. Such association between nNav1.5 and VEGF-A backs the findings reported by Andrikopoulos et al. [75] which indirectly connects back to the reverse mode of NCX. However, the direct principle of Nav1.5 as the driving force of NCX in breast cancer requires further investigation. Interestingly, Dulong et al. [72] brought up a prospect involving Nav1.5 and the electrochemical gradient. Dulong and colleagues [72] suggested that the Nav1.5-regulated RhoA protein may be partially discussed on the basis of the sodium current and intracellular calcium gradient. Nav1.5-conducted sodium currents may affect the electrochemical gradient of calcium because sodium influx activates voltage-dependent calcium channels and thus induces calcium entry. Indeed, the RhoA level has been shown to be regulated by cytosolic calcium concentration [80].

Throughout Section 2, it is quite prominent that MCF-7 and MDA-MB-231 were used side by side to highlight the role of Nav1.5 and nNav1.5 in promoting breast cancer metastatic traits such as migration and invasion. Both MCF-7 and MDA-MB-231 portrayed phenotypic and multiple genotypic differences [81]. MCF-7 is known as hormone-dependent, whereas MDA-MB-231 is referred to as TNBC. The lack of hormone receptors such as ER and progesterone receptors in TNBC cases has rendered insensitivity to anti-hormone-based treatments [82]. Therefore, it is important for investigators to identify potential therapeutic targets such as nNav1.5. Based on metabolic perspectives, MCF-7 cells tend to be described as the Pasteur type, which relies on the ATP synthesis from phosphorylation at normoxic conditions. However, their glycolytic activity is increased under hypoxic conditions. Unlike MCF-7, MDA-MB-231 cells are more likely to resemble the Warburg type, where the glycolysis for ATP synthesis takes place under both normoxic and hypoxic conditions [83,84]. In relevance to the context of nNav1.5, it was reported that the VGSCs-dependent invasiveness in colorectal cancer is majorly driven by nNav1.5 under both normoxic and hypoxic circumstances [85]. It was further demonstrated that the hypoxia-prompted increase in the invasiveness of colorectal cells is likely to be regulated by the persistent current component of nNav1.5 [85]. Aside from that, MCF-7 cells are known to express the epithelial phenotype, whereas MDA-MB-231 are more likely to portray mesenchymal-like phenotype [86].

In a study by Eiden and Ungefroren [86], the expression of Rac1B in MCF-7 is higher compared to MDA-MB-231. However, the protein levels of Rac1 are lower in MCF-7 but increased in MDA-MB-231 cells. Further investigation showed that the Rac1B/Rac1 ratio was 100% in MCF-7 and 0.83% in MDA-MB-231. The low Rac1B and high Rac1 were associated with the undifferentiated cells such as MDA-MB-231 and consistent with their high migratory feature [86]. Overall, the high Rac1B and Rac1 ratio may be necessary for maintaining epithelial differentiation and blocking the mesenchymal transition and malignant phenotype [86]. In relevance to Nav1.5, the activation of Rac1 by Nav1.5 has been reported to stimulate the motile phenotype and migration of breast cancer cells. Putting all these facts into perspective, we believe that MDA-MB-231 is a valid model to study for understanding the role of Nav1.5 in promoting metastases in breast cancer cells. 

In a comprehensive study by Shi et al. [87], the miRNAs of MCF-7 and MDA-MB-231 were compared. Of particular interest, we evaluated the findings from Shi et al. [87] based on MiR-21, miR-10b, and miR210. The expressions of miR-21, miR-10b, and miR-210 were higher in MDA-MB-231 compared to in MCF-7 [87]. MiR-21 is known to enhance invasion, migration, and EMT in breast cancer cells via various target genes [88,89]. The upregulation of miR-10b is closely related to the invasiveness of breast cancer and the acquisition of EMT [90,91]. MiR-210 was previously described as an independent prognostic factor of TNBC [92]. By reassembling this evidence, we found that the abilities of these miRNAs in mediating metastatic traits have also been found under the context of Nav1.5 and nNav1.5, as simplified in Figure 5. Such similarities further justify the relevance of pursuing the MDA-MB-231 cell line to study Nav1.5 and nNav1.5. 

## 3. The Connection between Breast Cancer and the Immune System

In 2011, the influential review by Hanahan and Weinberg [93] on the hallmarks of cancer was updated, whereby two more hallmarks of cancer were included. One of them was the evasion of immune destruction. The inclusion of the immunology perspective in understanding the progression of cancer is simply astounding and provides a more comprehensive interpretation of the disease.

The concept of metastasis is interconnected with the human immune system. During the metastatic cascade, cancer cells interact closely with the immune system and they influence each other, both within the tumour microenvironment and systemic circulation [94]. Thus, it can be said that the presence of migrating metastatic cancer cells may stimulate immune response within the human body [95,96,97]. The biomarkers present on these migrating cancer cells can be recognised by the immune cells and eventually trigger the immune system to produce antibodies to neutralise the expression of the biomarkers [95,98]. The crosstalk between cancer and immune cells contributes to another layer of complexity to our understanding of the formation of metastasis, but at the same time opens new therapeutic opportunities for patients such as cancer immunotherapy [96,99]. 

Breast cancer has traditionally been labelled as non-immunogenic or “cold”, meaning it does not provoke an immune response [21,100]. However, recent studies have proven otherwise. There are some astounding revelations that highlight the immunogenicity of breast cancer in various contexts [101,102]. One of those contexts is related to the heterogenicity of breast cancer. Breast cancer is made of a heterogeneous mixture of different molecular subtypes [21] and it has been reported that these subtypes contribute to the variation in the capabilities of inducing immune responses [103].

### 3.1. The Immunogenicity of Breast Cancer

A key insight is that the proportion of patients with tumour-infiltrating lymphocytes (TILs) varies depending on the hormonal receptor profile of the tumour. Substantial lymphocytic infiltration promises a more favourable prognosis [104,105]. TILs, found in these tumours, signify that the immune cells have recognised the tumour cells, multiplied, and infiltrated them to prevent the progression of the tumour. Patients with hormone receptor-positive tumours are unlikely to have TILs in their tumours [106], whereas the triple-negative subtype has a robust tumour T-cell infiltrate compared to any other subtypes [105]. On the other hand, HER2-positive breast cancer has also been shown to have a large number of patients with significant TILs in their tumour. In HER2-positive breast cancer, the numbers of TILs can be further elevated via the presentation of trastuzumab treatment [106,107,108].

As discussed throughout the review, the MDA-MB-231 cell line is vastly utilised to demonstrate the role of Nav1.5 and nNav1.5 in the context of breast cancer. Since the MDA-MB-231 cell line is a TNBC cell line, it is relevant for us to understand the basics of TNBC immunogenicity. 

A recent study reported that TNBC cases with a low homologous recombination deficiency (HRD) exhibited a greater level of CD8+ T lymphocytes as compared to hormone receptor-positive cancer with a high HRD [102]. The study emphasised that the levels of HRD are inversely correlated with the immunogenicity in primary BRCA 1/2 breast cancers [102].

According to several sources, TNBC (highly metastatic) is more likely to be infiltrated with TILs as compared to other subsets. This is because TNBC is known to be immunogenic due to the presence of genetic instability, increased gene mutations, and chromosomal instability that may end up encoding peptide epitopes that appear foreign to the immune system, thus provoking it to induce an anti-tumour response [105,106,109,110]. In an article by Loi [105], the author suggested that if an immune response can be developed to prevent the progression of TNBC, which can be determined based on the presence of TILs at the time of diagnosis, the prognosis may be improved independently of the use of chemotherapy. 

TNBC is also associated with another positive prognostic signature which is B cell-specific [111]. A gene expression study by Iglesia et al. [111] that focused on B cells indicated that the improved prognosis (metastasis-free survival) correlates well with B-cell gene expression, particularly in basal-like and HER2-enriched breast cancer subtypes. The presence of tumour-targeting B cells may elevate tumour inhibition via the production of antibodies (antibody-dependent cell-mediated cytotoxicity) that are initiated by natural killer cells (NK cells). The presence of multiple different peptide antigens on TNBC may initiate the production of antibodies which attracts numerous innate immune cells. These innate immune cells are capable of presenting the antigens to be recognised by T cells. The activation of adaptive immune response can be further enhanced with the presence of cytokines produced by NK cells that are recruited via antibody Fc receptors [106]. 

Aside from the mentioned markers, there are several other examples, such as LINC00460. In a study by Cisneros-Villanueva et al. [112], the expression of LINC00460 modulates various immunogenic-related genes in BRCA cancer such as SFRP5, FOSL1, IFNK, CSF2, DUSP7, and IL1A. Interestingly, the expression of LINC00460 was increased in BL2 type TNBC, and it potentially regulates the WNT differentiation pathway. Via the activation of the WNT/β-catenin signalling axis, cancer cells are capable of retarding anti-tumour effects by excluding CD8+ T lymphocytes [113]. Interestingly, Nav1.5 was said to be involved in the progression of oral squamous cell carcinoma via the activation of the WNT/β-catenin signalling pathway [114].

The improvement in the stratification of breast cancer subtypes via gene profiling and high throughput imaging has definitely contributed to the separation of strongly and weakly immunogenic breast cancer subtypes [115]. 

### 3.2. The Role of the Immune System in the Progression and Elimination of Breast Tumours

The immune system plays an important role in both tumour progression and elimination. The interaction between the immune system and developing cancer cells, also known as immunoediting, consists of 3 phases: elimination, equilibrium, and escape [116,117]. During the early stages of breast tumour development, the acute inflammatory response results in the production of IL-12 and IFN-gamma, thus establishing a Th type 1 environment at the tumour site [118,119,120]. During this phase, dendritic cells began to mature, process the tumour-associated antigens, and migrate to the tumour-draining lymph nodes to present the foreign antigen to naive CD4+ and CD8+ T cells [121,122]. As a result, the immune response ultimately reduces the number of tumour cells and completes tumour rejection. However, the pressure it imposes leads to the selection of tumour cell variants that tend to escape detection by the immune response [93]. 

The process of immunoediting establishes a state of equilibrium (otherwise known as dormancy) [21]. As inflammation at the tumour site shifts from acute to chronic, the tumour microenvironment evolves to a Th type 2 profile, and the immune cells found in the microenvironment of breast cancer majorly consist of type 2 cells. Cytokines such as interleukin IL-10 and IL-6 are expressed by CD4+ Th2 cells in order to reduce the destruction of the immune system [123,124,125]. Innate immune cells and high levels of regulatory T-cells, on the other hand, secrete substances that both dampen the function of CD8 T cells and prevent their migration to the tumour site, thus producing only a small number of less active CD8 T cells that are available in order to induce tumour regression. The most unique aspect of breast cancer immunology is that breast carcinoma is able to influence the function of antigen-presenting cells (APC) by secreting substances that manipulate T cells to become type 2 immune cells rather than type 1 [21,106,116]. 

Immune escape mechanisms of breast cancer can be correlated with the characteristics of the subsets of the disease. In hormone receptor-positive breast cancer, a low level of major histocompatibility complex class I (MHC class I) and absence of strong immunogenic tumour antigens contributes to the ability of the subset to survive unnoticed by the immune system. The presence of oestrogen (immunosuppressive) promotes tolerance of the weakly immunogenic cancer by polarising the immune system towards a type I immune response since most of the immune cells, including macrophages, T and B lymphocytes, and NK cells express ER [116].

In HER2-positive tumour cells, the presentation of MHC class I is inversely correlated with the expression of HER2 receptors [116,126]. In contrast, TNBC portrays a spectrum of MHC class I presentation and prominent tumour-associated antigen expression. Therefore, the concept of immune escape in TNBC focuses on the development of an immunosuppressed tumour microenvironment [127] that consists of immune checkpoint proteins such as programmed cell death protein 1 (PD-1)/programmed death-ligand 1 (PD-L1) [128,129] and other components such as T regulatory cells [130] and myeloid-derived immunosuppressor cells (MDSCs) [131] which supports the progression of this metastatic subtype.

Abnormalities in the MHC class I may contribute to the survival of cancer cells [132]. Pertaining to VGSC, a neuron study showed that the blockade of action potentials with Tetrodotoxin resulted in a solid increase in the mRNA that encodes for MHC class I [133]. Such a finding implies the possibilities of Nav1.5 and nNav1.5 in influencing the expression of MHC class I. 

The prospects of immune escape and the angiogenic switch are closely connected as well. Tumour-associated macrophages and other immune cells tend to release pro-angiogenic factors that can stimulate the angiogenic switch that contributes to the progression of tumour mass. There are two models proposed to explain this phenomenon. The first model suggests that once the tumour undergoes immune escape and the angiogenic switch is turned on, the tumour grows locally and metastases [116]. Disseminated tumour cells (DTCs) are released at a later stage because DTCs do not have access to the bloodstream until the tumour has acquired its own set of blood vessels or vasculature. The second model, however, proposes the opposite. This model stipulates that the dissemination of cancer cells may take place at the beginning stage of cancer development [134] and continues throughout its progression. This implies that the role of immune escape is more essential than the role of the angiogenic switch because microscopic or smaller-sized tumours may have spawned across the circulation even before the angiogenic switch is activated [116].

## 4. The Involvement of Nav1.5 in the Functionality of the Immune System and the Immunogenicity of nNav1.5

The natural immunogenicity of Nav1.5 is underexploited at the moment, and there is a limited source of literature available that links this sodium channel to breast cancer immunology. However, there are few studies that provide a fundamental understanding of the connection between Nav1.5 and the human immune system. These links were not directly associated with breast cancer immunology, but rather led us to understand the positive prospects of Nav1.5 in the context of immunology. Aside from that, this subsection highlights the immunogenicity of nNav1.5 [77]. 

As we have discussed earlier, the immune system can be divided into two main categories: the innate and adaptive immune systems. Macrophages are white blood cells that play essential roles in both innate and adaptive immune systems. The demonstration of Nav1.5 within phagosomes of activated macrophages [135] is undoubtedly a significant milestone in deciphering the noncanonical role of Nav1.5. It was reported that the expression of Nav1.5 within late endosomes of macrophages contributes to the regulation of the acidification of endosomal lumina by counterbalancing proton accumulation (Figure 6). Such electroneutrality provides a negative feedback loop that avoids injury to the cell. 

### 4.1. Nav1.5 and Immune-Mediated Diseases 

Furthermore, Nav1.5 has been studied in relation to immune-mediated diseases such as multiple sclerosis. Multiple sclerosis is caused by the damage of the myelin sheath, prompted by the immune system itself, where macrophages conduct the phagocytosis of the myelin. Such a circumstance contributes to the formation of myelin plaques. The regulation of phagocytosis [136] and myelin degradation are both dependent upon the acidification of endosomal lumina, which links back to Nav1.5. A study published in 2013 validated the hypothesis, reporting the involvement of Nav1.5 in the phagocytic route of myelin breakdown in macrophages within multiple sclerosis lesions [137].

### 4.2. The Role of Nav1.5 in Innate Antiviral Host Defense

In addition to immune-mediated disease, the role of Nav1.5 has been highlighted in the antiviral host defence as well [138]. It was suggested that the *SCN5A* expression (transcript) and channel activation in human macrophages may serve as central regulators of pathogen recognition and signalling pathways that are associated with antiviral host defence [138]. It was discovered that upon contact of the double-stranded RNA (pathogen-related molecular arrangement), the *SCN5A* channel variant in macrophages becomes activated, which leads to the stimulation of the cyclic AMP (cAMP) signalling pathway. The stimulation of the cAMP pathway plays an essential role in the transcription of activating the transcription factor-2 (*ATF-2*) gene. With the assistance of the *ATF-2* gene transcription, *SCN5A* expression in macrophages controls antiviral-associated genes [138].

### 4.3. The Role of Nav1.5 in Adaptive Immune System: T-Lymphocytes

The adaptive immune system is a much more complicated scene as compared to the innate immune system. The adaptive immune system involves the presence of T- and B- lymphocytes and the production of antibodies. Antigen presentation by MHC protein is the driving force in adaptive immunity [139]. The immature form of T-lymphocytes are known as thymocytes. Thymocytes undergo a screening process in the thymus where they are subjected to non-, positive, and negative selections. The positive selection highlights a mild interaction between the T-cell antigen receptor (TCR), self-peptide, and MHC. Such positive selection is necessary to rescue the double-positive thymocytes that express TCR from death by negligence. The influx of Ca^2+^ into CD4+CD8+ double-positive thymocytes is crucial to regulating the positive selection [140]. A ground-breaking study in 2012 reported the influence of Nav1.5 (*SCN5A*) on the positive selection of CD4+ T-lymphocytes by sustaining the influx of Ca^2+^ ions [140]. Furthermore, the predominant presence of Nav1.5 has been reported in Jurkat cells which exhibit functional characteristics that mimic those of the T-lymphocytes [141]. The association between VGSC and lymphocytes has been prompted by the earlier findings published by Gáspár Jr et al. [142] and Pieri et al. [143].

### 4.4. The Immunotherapeutic Prospects of Nav1.5 and nNav1.5: Cancer Immunotherapy and Immune Evasion

Cancer immunotherapy is an emerging branch in the field of medicine. There are several spectrums that are classified under breast cancer immunotherapy which includes the introduction of cytokines [144,145,146], cell-based therapies: vaccines [147] and adoptive cellular therapy [148,149,150], immune checkpoint blockade [151,152], and combination immunotherapy [145,153]. 

Therapeutic antibody targeting or immune checkpoint blockade is a method whereby the exclusive antigen of a cancerous cell is exploited to prevent its role in potentiating cancer development [154]. Recently, in March 2019, the U.S. Food and Drug Administration (FDA) [155] has approved an immune checkpoint inhibitor known as atezolizumab (Tentriq) in order to treat patients with metastatic TNBC. This is a result of the IMpassion130 trial, which was a randomised phase III clinical trial designed to test whether the addition of atezolizumab to standard chemotherapy could improve outcomes for patients with metastatic TNBC. TNBC is the breast cancer subtype most associated with the upregulation of PD-L1 [156,157]. The combination is approved for women who are diagnosed with locally advanced or metastatic TNBC that cannot be treated surgically and whose tumours are positive for PD-L1 [158]. Atezolizumab is a monoclonal antibody that binds to PD-L1, blocking its binding to and activation of PD-1, which is expressed on activated T-cells, that may enhance the T-cell-mediated immune response towards the tumour and reverse T-cell inactivation [159]. 

The immunotherapeutic use of antibodies against Nav1.5 and nNav1.5 has been reported to be beneficial in preventing the progression of breast cancer [18,160]. In 2005, Chioni et al. [161] introduced an anti-peptide polyclonal antibody, named NESOpAb (sequence: NH2-VSENIKLGNLSALRC-amide), which specifically recognised the ‘neonatal’ but not ‘adult’ Nav1.5 when tested on cells that specifically over-express either one of these Nav1.5 spliced forms (Figure 7). The antibody was used to investigate the developmental expression of nNav1.5 in a range of mouse tissues via immunohistochemistry. Interestingly, it was demonstrated that the NESOpAb blocked the functional nNav1.5 ion conductance in human embryonic kidney (EBNA-293) cells when applied extracellularly at concentrations as low as 0.05 ng/mL. Aside from that, NESOpAb was also incorporated in a study where the introduction of NESOpAb led to the prevention of migration and invasion in MDA-MB-231 cells [18].

In 2019, Gao and colleagues [160] exploited the third extracellular region (S5/S6) of Nav1.5 to prevent the progression of three types of cancer; breast cancer (breast adenocarcinoma cells, MDA-MB-231), ovarian cancer (ovarian adenocarcinoma cells, Caov-3), and cervical squamous cancer (SiHa cells). This study incorporated the designation of the Nav1.5-third extracellular region antibody (E3Ab) to specifically bind to and inhibit Nav1.5. The antibody was raised in a rabbit model that had been immunised against a synthetic 18 amino acid peptide (CVRNFTALNGTNGSVEAD) based on the coding sequences of D1:S5/S6 (Figure 7). E3Ab is a peptide-specific polyclonal antibody that recognises E3 of Nav1.5 [160]. The application of E3Ab was previously shown to inhibit about 60% of Na+ current in HEK-293 and Chinese hamster ovary epithelial cells after an extracellular application [162]. Such a finding is relevant to the site at which the E3Ab of Nav1.5 targets, which is the pore (D1:S5/S6) where the current is conducted as illustrated in Figure 1 and Figure 7. 

Gao and colleagues [160] found that the cancer cells that were pre-treated with E3Ab had a significantly attenuated invasion ability as compared with the control group. This is demonstrated by the decreased number of cells that could migrate across the Matrigel filters. The migration ability of MDA-MB-231 cells that had been treated with E3Ab was decreased by 21.2% as compared with the negative control [160]. 

We believe that the E3Ab antibody decreasing the migration of cancer cells is associated with the protein expression of MMP-9. In the study by Gao et al. [160], migration was reduced in the cancer cells after the introduction of E3Ab, and the protein expression of MMP-9 was reduced by E3Ab. Previous studies have demonstrated that MMP-9 plays an important role in the migration of cancer cells such as MDA-MB-231 [163,164]. Interestingly, in a previous study, Gao et al. [165] showed that the blockage of Nav1.5 channel in MDA-MB-231 reflected a decrease in MMP-9. In agreement, Mohammed et al. [47] also demonstrated that the knockdown of Nav1.5 resulted in the prevention of invasion in endocrine-resistant breast cancer cells via the decline in total MMP activity. By putting all these findings into perspective, it is believed that the E3Ab decreases migration in cancer cells by interfering with the role of MMP-9 in the context of migration.

### 4.5. The Natural Immunogenicity of nNav1.5

The amino acid sequence in the extracellular loop in VSD1 (D1:S3–S4) of nNaV1.5 is unique and would be expected to be highly antigenic [166]. Recently, the natural immunogenicity of nNav1.5 was reported via the detection of circulating antibodies against nNav1.5 (hereafter referred to as anti-nNav1.5-Ab) [77]. The production of anti-nNav1.5-Ab was elevated in the pre-treated breast cancer patients as compared to those who received treatments. This is in line with the progression metastasis in untreated breast cancer patients. However, the breast cancer patients who have undergone treatment exhibited the downregulation of anti-nNav1.5-Ab, which reflects the reduced serum level of nNav1.5. It has previously been suggested that the employment of treatment such as chemotherapy and radiation in breast cancer may stimulate the expression of MHC class I [167]. Such phenomenon is supported via findings discovered by Murthadha et al. [168], where the downregulation of nNav1.5 in aggressive breast cancer cells positively upregulates the expression of MHC class I. Such upregulation reflects the activation of the antitumour immune response, which is crucial in eliminating oncogenic proteins, including nNav1.5. In addition, the fluctuating serum level of anti-nNav1.5-Ab in response to breast cancer treatment could also suggest the capability of these novel antibodies to act as immunosurveillance markers to track the progression of breast cancer and monitor treatment effectiveness [77]. 

## 5. Reassembling the Triad and Future Perspectives

In the earlier subsections, we have analysed the triad in three dimensions, encompassing Nav1.5 (nNav1.5), breast cancer, and the immune system. Once the triad has been reassembled, we were able to derive several sub-concepts (Figure 8) that may play crucial parts in the overall concept of breast cancer immunotherapy.

### 5.1. Sub-Concept 1: Nav1.5 and nNav1.5 as Immunotherapeutic Targets in Combatting Breast Cancer

Nav1.5 and nNav1.5 have been known to potentiate the metastatic cascade which is necessary for breast cancer progression [18,49,70]. In terms of the immune system functionality, Nav1.5 plays an important role in the positive selection of CD4+ T-lymphocytes [140], whilst the immunogenicity of nNav1.5 validates its vulnerability towards immune responses [77]. In assembling this evidence, we propose that both Nav1.5 and nNav1.5 could play crucial parts in breast cancer immunotherapy. The positive selection of T-lymphocytes by Nav1.5 [140] could be used as a tool to increase the recruitment of CD4+ cells to reverse immunosuppression in breast cancer [169]. Since the immunogenicity of nNav1.5 is proven, the neonatal channel can be used as an immune checkpoint marker, such as in the case of PD-L1. Commercial production of antibodies against Nav1.5 [160] and nNav1.5 [161] could be incorporated in clinical studies to decipher the immunotherapeutic abilities of Nav1.5 and nNav.5 in combatting breast cancer progression. 

Additionally, there is a lack of evidence to support the explanation surrounding the antigen processing machineries involving Nav1.5 and nNav1.5. This could also be an interesting prospect to study in the future. 

### 5.2. Sub-Concept 2: Nav1.5 and nNav1.5 as Immunotherapeutic Targets in Combatting TNBC

The externalisation of Nav1.5 [64] in the plasma membrane of the TNBC cell line and the inverse relationship between ER status and nNav1.5 [13] have been demonstrated. It was emphasised that such externalisation contributes to the Na+ current, which is responsible for assisting breast cancer progression. This is definitely apt in the case of TNBC, as it does not express ER/PR/HER2, which increases the efficiency in the functional roles of Nav1.5 and nNav1.5 in promoting breast cancer progression. The extensive use of MDA-MB-231 in demonstrating the actions of Nav1.5 and nNav1.5 has sealed the connection of these sodium subunits and the progression of the TNBC subtype.

From an immunological perspective, TNBC is an immunogenic breast cancer subtype that exhibits increased infiltration of TILs [170]. However, there is no direct evidence that associates the natural immunogenicity of nNav1.5 and TNBC subtypes in clinical settings. Thus, the future perspectives here would be: (a) does the immunogenicity of nNav1.5 increase in the cases of TNBC and (b) does the level of anti-nNav1.5-Ab increase in the serum of TNBC breast cancer patients? 

### 5.3. Sub-Concept 3: Blocking nNav1.5 to Increase MHC Class I Expression in Breast Cancer Immunotherapy for TNBC Patients

Currently, atezolizumab is provided for TNBC patients who expressed PD-L1 in their tumours. The prescription of atezolizumab highlights the incorporation of immunotherapy to assist conventional breast cancer treatments. In addition to infiltration of TILs, TNBC also possesses a low level of MHC class I proteins which is crucial for the survival of the subtype [171,172]. Since the downregulation of nNav1.5 rescues the expression of MHC class I [168], we believe that by targeting nNav1.5 using compatible immune checkpoint inhibitors, the progression of TNBC could be suppressed.

### 5.4. Sub-Concept 4: Role of Nav1.5 in Promoting Immune Evasion via Angiogenic Signalling Axis

The involvement of Nav1.5 in angiogenesis has been demonstrated by Andrikopoulos et al. [75]. Since there is an established ‘bridge’ connecting angiogenesis and immune evasion [173], we believe that it is possible that Nav1.5 (or nNav1.5) may modulate immune evasion by altering the angiogenic signalling axis.

Recently, Rajaratinam et al. [77] pointed out the upregulation of IL-6 and the significant positive correlation between anti-nNav1.5-Ab and IL-6 in the serum of breast cancer patients who did not receive treatment. This implies that the progression of metastasis is supported by the upregulation of IL-6 [77]. What is so unique about IL-6 is that it does not only promote breast cancer metastasis [174,175] and angiogenesis [176,177] but also contributes to immunosuppression [178]. Putting things into perspective, we believe that nNav1.5 might work along with immunosuppressive agents such as IL-6 and manipulate the angiogenic signalling axis in order to assist immune evasion in breast cancer. 

Adopting the concept of immune evasion proposed by Hanahan and Weinberg [93] in their updated version of ‘Hallmarks of Cancer: The Next Generation’, we believe that the potential of Nav1.5 and nNav1.5 in evading the immune system remains underexploited. Numerous gaps remain unanswered when incorporating Nav1.5 and nNav1.5 into the notion of immune evasions pertaining to breast cancer and breast cancer immunotherapy, such as a) how does Nav1.5 (or nNav1.5) assist in immune evasion to rescue to the progression of breast cancer, b) does the expression of Nav1.5 influence angiogenesis, thus indirectly regulating immune evasion, and c) is there a link between nNav1.5 and Il-6 when it comes to immune evasion in breast cancer?

## 6. Limitation of the Review

As for the limitations of this review, we believe that there is a lack of information on the direct association between Nav1.5/nNav1.5 and breast cancer immunology. The lack of information may include the absence of investigation on the immunological pathways involved in the processing of Nav1.5 and nNav1.5. Additionally, there is a limited number of literature sources available on the in vivo investigations pertaining to nNav1.5 and the immunological areas which consist of MHC classes, macrophages, and regulatory T-cells under the context of nNav1.5. 

## 7. Concluding Remarks

Throughout the review paper, we have discussed the various dimensions of the triad encompassing three main components which are Nav1.5 (nNav1.5), breast cancer, and the immune system. In a nutshell, the upregulation of Nav1.5, especially in its neonatal form, potentiates breast cancer metastasis. Since there is proven crosstalk between breast cancer metastasis and the immune system, the immunogenicity of nNav1.5 can be manipulated as a potential immunosurveillance marker to detect the progression of breast cancer metastasis. The involvement of Nav1.5 and its neonatal isoform in the immune system’s functionality is definitely a beneficial scope of study that requires more attention. Overall, the triad has contributed several valuable sub-concepts that could ultimately assist breast cancer immunotherapies involving Nav1.5 and nNav1.5 as their star casts. 

## Figures and Tables

**Figure 1 biomolecules-12-00310-f001:**
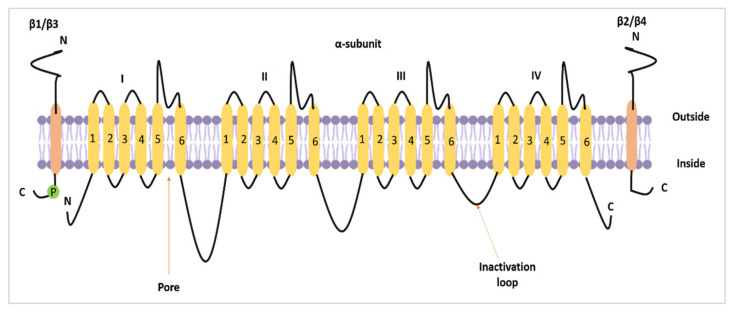
The topology of a VGSC α subunit and β subunits. The figure was adapted and modified from Brackenbury and Isom [4] and recreated.

**Figure 2 biomolecules-12-00310-f002:**
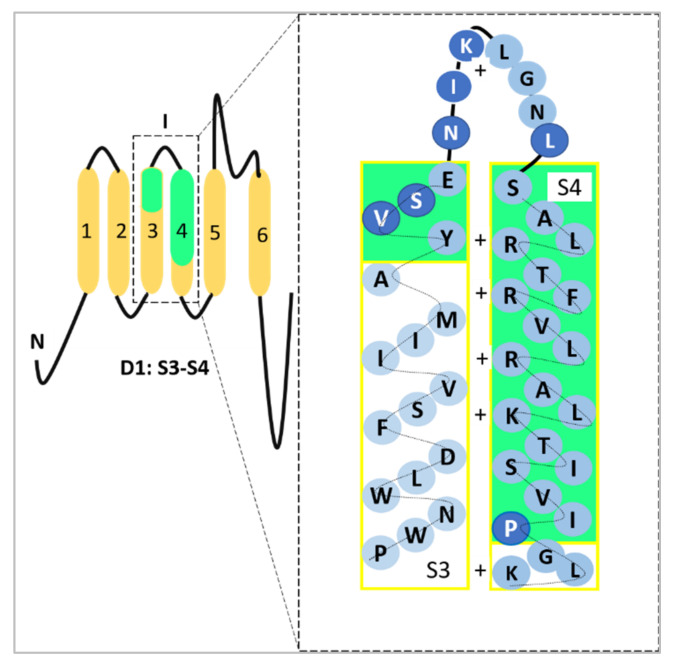
A representation of the site where the D1:S3 splicing of Nav1.5 takes place. The seven amino acid difference has been highlighted in darker blue circles which could be observed in D1:S3–S4 and within the S3/S4 linker. The D1:S3–S4 sections have been highlighted in green to emphasise the exact location which represents nNav1.5. The four positively charged residues in S4 (voltage-sensor) were indicated as well. This image was adapted and modified from Onkal et al. [12] and recreated.

**Figure 3 biomolecules-12-00310-f003:**
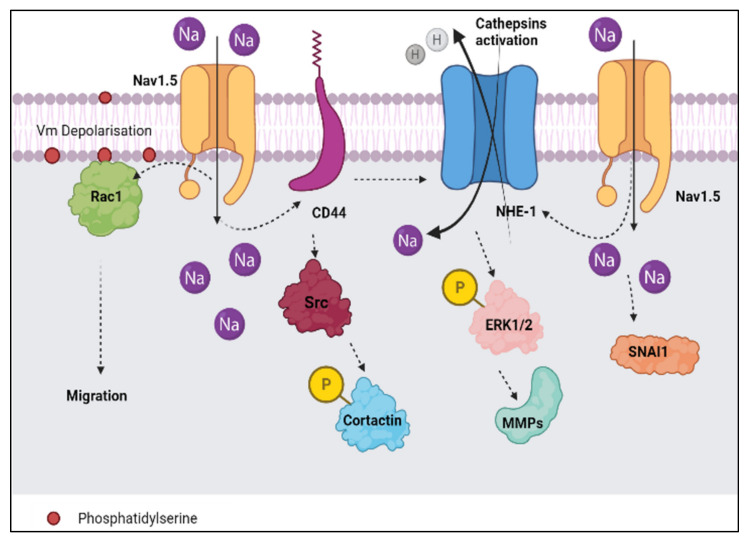
The influx of sodium ions (represented by the solid arrow in Nav1.5) leads to the efflux of hydrogen ions or protons (represented by the solid arrows in NHE-1) into the extracellular surrounding. Nav1.5 interacts with NHE-1 in the focal ECM degradation sites and enhances the invadopodial proteolytic activity contributing to breast cancer invasion. Additionally, the figure summarises the involvement of Rac1, CD44, Src, ERK1/2, MMPs, and SNAI1 under the context of Nav1.5 (represented by the dashed lines). The Vm depolarisation may regulate Rac1 activation via its interaction with phosphatidylserine. The figure was adapted and modified from Luo et al. [22] and recreated using BioRender.com (accessed on 8 January 2022).

**Figure 4 biomolecules-12-00310-f004:**
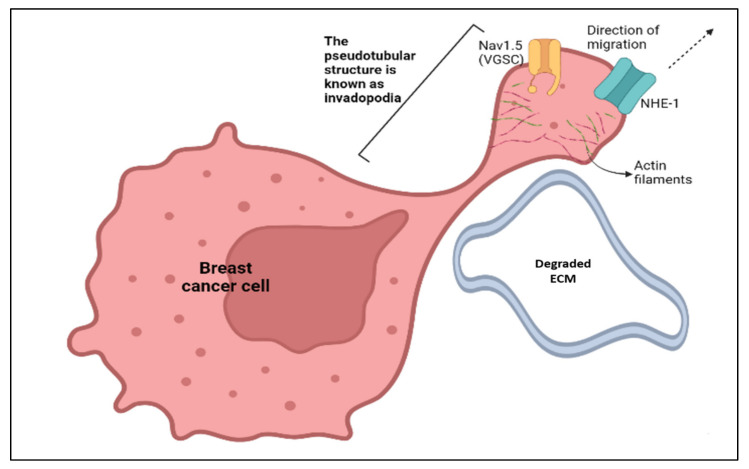
The invadopodial activity pursued by NHE-1 and Nav1.5 under the context of breast cancer cell migration. The figure was created using BioRender.com.

**Figure 5 biomolecules-12-00310-f005:**
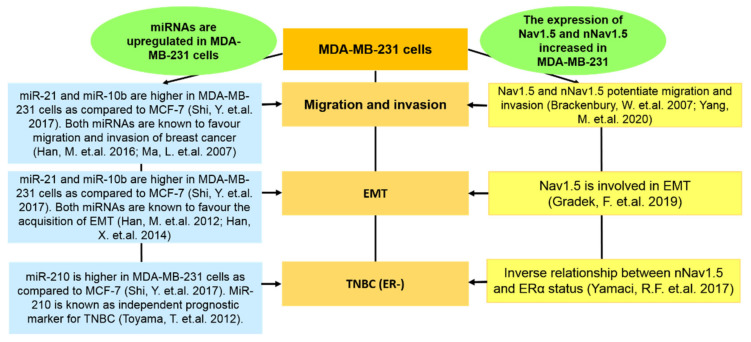
The similarities in the metastatic traits pursued by miRNAs and Nav1.5 (nNav1.5) under the context of MDA-MB-231 cells. As suggested by Shi et al. [87], the upregulated expressions of miR-21, miR-10b and miR-210 are higher in MDA-MB-231 cells. Both miR-21 and miR-10b are known to favour migration, invasion and EMT [88,89,90,91] and such traits can be observed in the contexts of Nav1.5 and nNav1.5 as well [18,65,70]. Additionally, the expressions of both miR-210 [92] and nNav1.5 [13] are closely related to TNBC.

**Figure 6 biomolecules-12-00310-f006:**
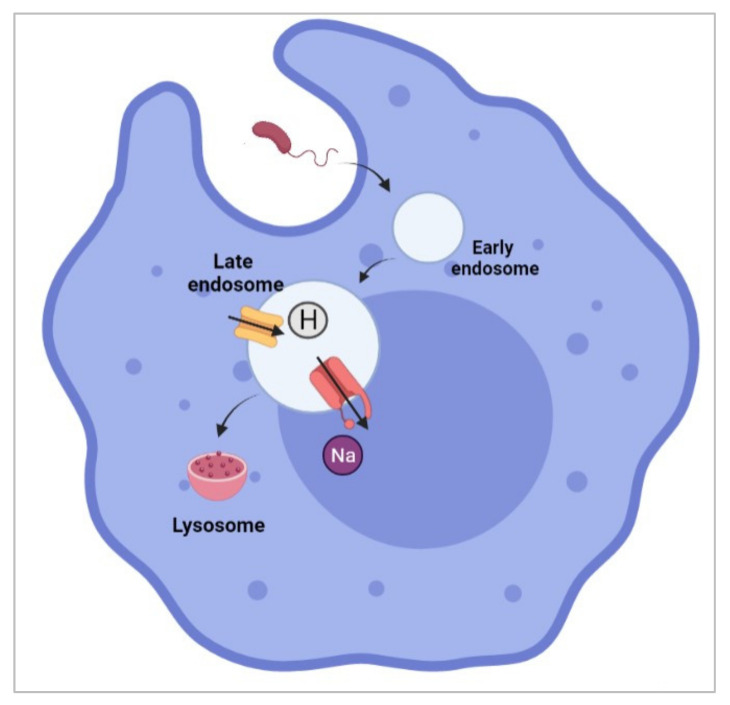
Nav1.5 channel facilitates sodium efflux which promotes endosomal acidification (proton accumulation) and phagocytosis within the late endosomes of the macrophage. This image was created using BioRender.com.

**Figure 7 biomolecules-12-00310-f007:**
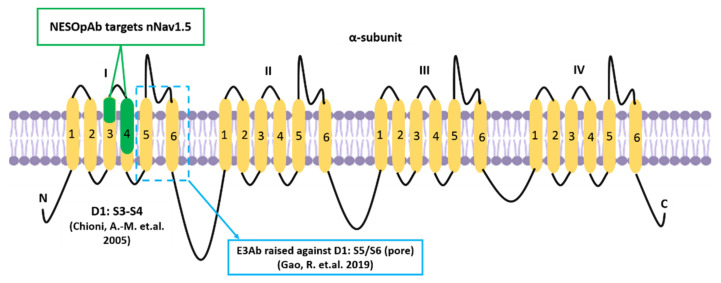
The previous illustration (Figure 1) was modified to mark the locations at which NESOpAb and E3Ab target. According to Chioni et al. [161] and Gao et al. [160], NESOpAb targets the D1:S3-S4 region whereas E3AB targets against D1:S5/S6 region, respectively.

**Figure 8 biomolecules-12-00310-f008:**
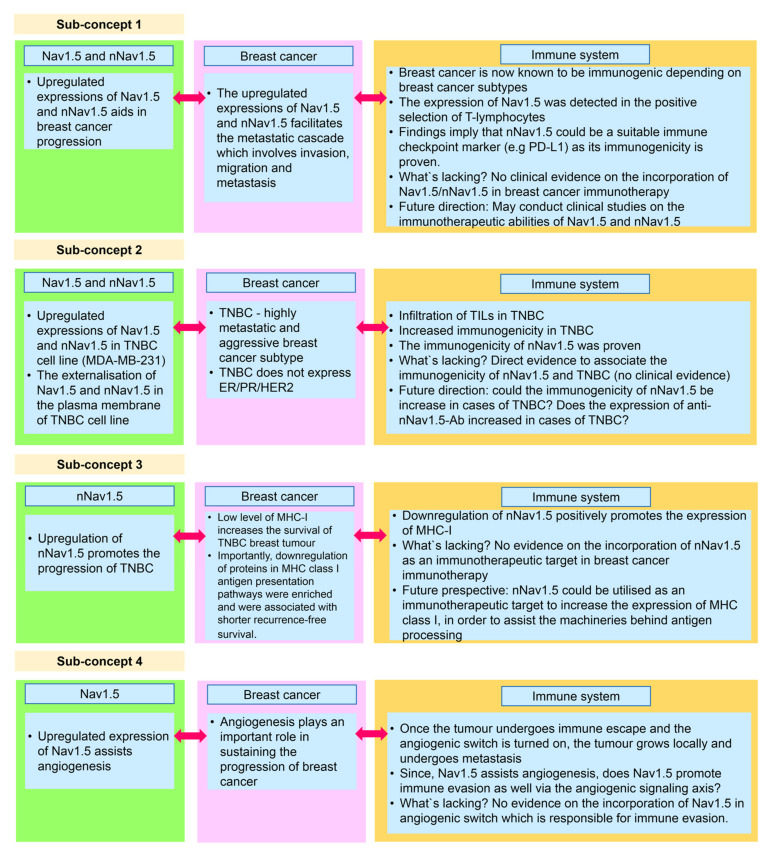
The summary of the sub-concepts that are derived from the triad.

## Data Availability

Data sharing is not applicable to this article as no datasets were generated or analysed during the current study.

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
