# Peer review of "Discovering the Triad between Nav1.5, Breast Cancer, and the Immune System: A Fundamental Review and Future Perspectives"

_biomolecules, 2022, doi:10.3390/biom12020310_

Round 1

Reviewer 1 Report

In general it is a good review of a specialised area within the fields of voltage-gated sodium channels, cancer and immune function. As the authors point out there is little out there at present on these subjects as a collective and this provides a good narrative and illustrates some interesting links between the three areas. The authors have discussed the different splice variants of Nav1.5 and their emerging roles in breast cancer and the immune system, considering some possible non-canonical signalling roles. Though in places further clarification of the role of Nav1.5 could be elaborated on. This is a rather timely review given the progress in the relative subject matters and one could imagine there will be much progress to come, which will allow for a deeper investigation. However, there are a few issues that need to be addressed as well as some possible scope for clarity in some areas.

In the discussion of pharmacological approaches to target Nav1.5 and nNav1.5 it would be of benefit somewhere to mention the potential pitfalls of targeting Nav1.5 and the relative merits of nNav1.5 over Nav1.5 (i.e that nNav1.5 is not endogenous within adult tissues). Is there any difference in the incidence of Nav1.5 and nNav1.5 in breast cancer – if nNav1.5 is more prevalent this would potentially be more amenable to reduced side effects of therapy, assuming it could be selectively targeted over Nav1.5. Additionally, how are these antibodies which inhibit Nav1.5/nNav1.5 in cancer having a beneficial effect? What is the mechanism? Is it through effects on signal transduction or gating behaviour of the channels? Or some other mode?

There is a lot of switching between Nav1.5 and nNav1.5 and in the context of discussing the different splice variants and their functional activity, the review would benefit from a structural figure to highlight the sequence differences (and any potential functional differences) as well as where antibodies mentioned would bind. For example in lines 61 – 68 the domain segments mentioned will be hard to follow for a non-specialist of VGSCs.

Specific issues.

Introduction

There are some fundamental issues with the background material on voltage-gated sodium channels – the authors may want to refer to some more specialised reviews to help clarify these. In particular, in lines 32 – 35, the authors refer to 4 distinct conformational states, however, there are technically 3. The closed/resting state, the open/activated state and the closed/inactive state. It is the latter of these that encompasses the refractory period where the channel is inexcitable to a depolarising stimulus. Numerous references to this can be found including Ahern CA et al., 2016 JGP and Catterall et al., 2020 Nature Chemical Biology.

Similarly, the authors state that Nav1.5, Nav1.8 and Nav1.9 are categorised as fast-inactivating channels, whereas the rest are known as the slow inactivating channels. This is simply not true. All mammalian VGSCs possess fast inactivation, it is fundamental property of VGSCs that is crucial for control of cell excitability in neurons and muscle, and for repetitive firing in neurons. Fast and slow inactivation are two separate processes which can occur within all mammalian VGSCs and are not categorical of subtypes. Invertebrate VGSCs on the other hand can be referred to as slow inactivating and cannot undergo the fast inactivation gating process (there are specific structural differences between these families of VGSCs). The three Nav channels mentioned can be distinguished from the remaining Nav channels on the basis of evolutionary divergence and loss of TTX sensitivity (relatively). VGSCs can also vary in their gating kinetics and voltage sensitivities, but these are subtle parameters that are very different to the concepts of fast and slow inactivation.

In the following section, despite it being an interesting topic, it is not clear to me why so much detail has been included on the mechanism of TTX sensitivity, given there is no apparent reference to this within the context of breast cancer and the immune system. This review focuses on Nav1.5 and so the differing properties of the various Nav alpha members does not necessitate this. The introduction would benefit from a more general overview of the VGSC structure in relation to its function both classical and non-canonical, the latter of which is the primary focus of this review given that the expression is in non-excitable tissues.

Section 2

Is there any hypothesis as to how omega 3 downregulates nNav1.5 expression?

Lines 163 – 165. This sentence is not clear. Was siRNA for Nav1.5 and nNav.1.5 introduced? Or was Nav1.5 and nNav1.5 protein introduced. If the latter, what was the siRNA?

Line 209 -210. Is it necessary to say ‘as compared to the one by Fraser et al. (2005)’

Paragraph at line 216. Introduction of ER status without definition of ER and ERα. As the sodium channel comprises an alpha subunit too, this does warrant clarification.

In this section there is reference to Nav1.5 being expressed at either the plasma membrane or in the cytoplasm. It is not clear where in the cytoplasm this Nav1.5 is being expressed, but presumably it is membrane bound within some internal organelle and not freely floating (as suggested by Fig 1).

Figure 1 contains rather a lot of text. Would it be possible to simplify this to convey the message largely pictorially?

Lines 242 – 246. The conclusion that expression of nNav1.5 and hormone receptors is mutually exclusive is reasonable on the basis of the evidence. Could the authors clarify what they think contributes to this, or at the very least comment on the lack of information here.

Line 255 – 256. Do the authors mean extracellular membrane or extracellular matrix?

Figure 2. Firstly, the sodium channel is inserted into the membrane the wrong way round. Na+ ions do not travel back through the channel. Secondly, this is quite a limited figure in terms of illustrative purposes and would benefit from some inclusion of either what drives these processes and/or how the change in H+ within the ECM leads to its degradation and the subsequent functional consequences for breast cancer progression.

Paragraph at lines 329 -335. Increased expression of RhoA is associated with metastasis and its silencing is reported to downregulate expression of Nav1.5 mRNA as well as the protein function (current). It then goes on to state that downregulating Nav1.5 decreased protein level of RhoA. Was this Nav1.5 inhibition at the protein or mRNA level? Does this suggest that inhibiting either one of the transcripts results in a functional knockdown of the other? i.e. it is not a sequential activation pathway but they are interdependent? Further clarification here is needed.

Lines 380-381. Interestingly there appears to be a role for reverse mode function of the NCX, which would act to increase intracellular Ca2+. Normally, the predominant function of the NCX would be forward mode in order to keep cytoplasmic levels of Ca2+ maintained, unless the membrane potential exceeds -20 mV (Baartscheer A et al. Front Physiol. 2011). What is the driving force for this reverse mode function? Perhaps, an increase of Na+ entry through Nav1.5, but this is not clarified.  

Section 3

Line 481. Regulatory T cells are abbreviated as Treg but then subsequently are referred to as regulatory t cells, is there any need to define as Treg if it is not going to be used.

Section 4

Line 573. The activation of SCN5A in macrophages is ambiguous. Are the authors referring to the SCN5A transcript? Activation of the functional channel in classical terms of voltage-gating?

Lines 601 – 605. This comes back to the earlier point about how a diagram would be useful to highlight the differences between Nav1.5 and nNav1.5 – here it is particularly interesting as you have an antibody specific to one form.

Similarly, line 611 - 619, the third extracellular region is not clear. Is this domain III? The third extracellular loop of DI? A diagram would really help. There is also mention of another antibody.  How does this antibody decrease migration of cancer cells?

Line 624. The extracellular loop in VSD1 of nNav1.5. Which extracellular loop? The VSD technically refers to S1-S4 of each domain (and S5 – S6 is the pore forming domain), so do the authors mean the S1-S2 loop or the S3-S4 loop?

Section 5

Line 691-692. This is a little repetitive. Introduction of VEGF and explanation of what it does should really have come with the first mention of VEGF in section 2.

Language

There are many instances of missing articles and inappropriate use of tenses which need to be rectified.

Examples:

Line 56. The needed before Nav1.5

Line 99. The needed between by and international.

Line 143. Almost and over are not both needed.

Line 188. An needed between using and animal.

Line 218. To needed between on and the.

Line 244. The needed between On and contrary.

I will not list all, but there are lots more incidences of these omissions throughout the manuscript which the authors should address.

Line 546. ‘was only discovered till early last year’ does not make sense.

Line 586. ‘T lymphocytes is known’ should be ‘T lymphocytes are known’

Line 674. ‘However, there are no direct’ should be ‘However, there is no direct’

Lines 677 and 678 increased should be increase

Again, the authors should check the manuscript thoroughly for these errors.

Figure 4. Sub concept 1. 3rd box. 3rd bullet point ‘a’ is missing between ‘could be’ and ‘suitable immune checkpoint’ and markers should be marker.                 Sub concept 2. Box 3. Last bullet point, ‘be’ is missing and ‘increased’ should be ‘increase’.                     Sub concept 3. Box 3. First bullet point, ‘of’ missing after downregulation. Second bullet point, ‘an’ missing after as and before immunotherapeutic.                        Sub concept 4. Box 3. First bullet point, is ‘metastases’ plural correct? Should it be undergoes metastasis?  

Author Response

Dear reviewer 1, please have a look at the attachment. Thank you for your assistance and guidance. 

Reviewer 2 Report

The authors present an interesting review of Nav1.5 voltage-gated sodium channel-alpha subunit and it’s role in breast cancer and relationship to immune characteristics. I have the following  comments and recommendations:

The paragraph in lines 104-116 appears to ignore many basic principles of breast cancer. For example, the statement in line 107-108 that LCIS is a breast cancer is incorrect. It is not a breast cancer, it is not a premalignant lesion and does not develop into breast cancer. It is a marker indicating an increased risk for the development of breast cancer. The "newer" classification systems which are described are not in clinical use. The statement in lines 114-116 is incorrect. For example, the distinction between mucinous carcinoma and triple negative breast cancer are made histologically, have different prognoses, and indicate different treatments. These statements need to be corrected.

The finding that the cellular distribution of nNav1.5 is related to ER presence is interesting. The sentence in line 245 suggests these may be causally related. Is there any evidence foe a hormonal relationship to the distribution of nNav1.5?

Many studies are cited analyzing two breast cancer cell lines, MDA-MB-231 and MCF-7. Both of these cell lines were developed from malignant pleural effusions from women with breast cancer, and thus are metastatic disease with a poor prognosis. However, the former is referred to as "highly metastatic breast cancer cells", and the latter as "weakly aggressive cell". Intuitively, why should the comparison of two malignant cell lines whose clinical outcome was similar, but which have different molecule characteristics, be a valid model for understanding the mechanisms of metastases. I recognize that these cell lines have been used extensively in publications to study Nav1.5. It would therefore be useful to describe briefly the characteristics of this model and the justification for their use, as you refer to them throughout this review.

Line 159, reference 33 apparently is an abstract that I was not able to retrieve. What were the findings for the normal cell line MCF10A?

The section on mechanisms , lines 248 – 375, is very good.

The section on the immune system in breast cancer is long, and it is difficult to do justice in a few paragraphs to a subject that is this expansive and complex. Importantly, this section does not add to our understanding of the subsequent sections. I would delete this section, and incorporate the needed concepts into the subsequent sections as they are required.

The summary section and Figure 4 is very good.

I would include comments on the limitations of the review.

Author Response

Dear reviewer 2, please have a look at the attachment. Thank you for your assistance and guidance. 

Round 2

Reviewer 1 Report

The authors have addressed most of my comments and the changes implemented have largely strengthened their narrative and the manuscript. However, there are still a few points that need clarifying or correcting.

Line 39. ‘Refractory is a period of recovery from inactivation whereby the channel cannot be opened in response to depolarisation’. This is poorly worded.

Line 45 – 46. Resistance to molar concentrations of TTX is referred to – this is not accurate, concentrations in the M range would inhibit all sodium channels including the TTX insensitive ones such as Nav1.5! (the authors later mention use of 30 µM TTX as a concentration used on Nav1.5, which is in direct contrast with suggestions that concentrations in excess of the M range are needed)

Lines 49 – 53. The authors state that the insensitivity of Nav1.5 to TTX is highlighted by a study where in a breast cancer cell line Nav1.5 was knocked down and subsequent 30 µM TTX treatment did not exhibit any further effect.

If Nav1.5 has been knocked down, or knocked out, then the current being targeted has already been removed/inhibited, what is TTX meant to do here? I don't see how this provides evidence for the argument that Nav1.5 is TTX resistant. I am still a little unsure why so much detail is being provided on TTX? However, the study referenced is interesting in the context of knockdown of Nav1.5 in this cell line and a reduction of cell invasion. The authors then mention that TTX decreased the level of pY421-cortactin, what is the context/relevance of this? Is TTX more important for this rather than for regulation of Nav1.5? If important, perhaps it is something that should be discussed later in the manuscript, rather than in the introduction of Nav1.5 (?)

Lines 56 – 60. This paragraph is a bit disjointed from the prior and following. Could it be moved to follow line 40? This way all the functional description of the channel is out of the way prior to moving on to channel distinction by TTX and alternative splicing.

Line 141. Should it be ‘were defined’ or ‘are defined’, if the former, has the definition for TNBC changed?

Line 193 – 194. What is the implication of the statement here? And what is TTX doing? Does increased sensitivity to nanomolar TTX upon nNav1.5 silencing mean there is another TTX-sensitive voltage-gated sodium channel present? Or is the cell sensitive to TTX in some other way than reduced Na+ current? Without clarification this does not elaborate on the role of TTX sensitivity in the context of breast cancer.

Paragraph at line 195. The study by Erdogan and Ozpolat is discussed in some detail, yet the findings are consistent with the studies of Roger et al., Isbilen et al. and Brackenbury et al in the two previous paragraphs. Could these sections not be condensed to summarise the key points rather than repeating information?

Line 217-219 What is the evidence that Phenytoin downregulates the expression of nNav1.5? I don’t think either paper referenced claims this.

Line 268. What does ‘the organelle membrane’ mean? Is a specific organelle being referred to? In fig 3. It is being depicted as the endoplasmic reticulum – so is this actually expressed there as it’s functional location? Or is it just during the protein folding and trafficking process? Normally, in the ER as the protein is being made and folded it would be inserted into the membrane so that the intracellular portion of the channel is still within the intracellular/cytoplasmic compartment of the cell and the extracellular face would sit inside the ER. (The use of a schematic without the 'ball and chain' could be used to prevent this directionality confusion - though this is totally up to the authors choice)

Line 691. Use of ‘SCN5A’ for the first time. Throughout the paper even when referring to silencing by siRNA the protein name (Nav1.5) has always been used rather than the gene. Why now the sudden switch? If necessary, it is important to make it clear that SCN5A encodes for Nav1.5 or some readers might be confused.

Minor comments/typos:

Line 52 – short hairpin Nav1.5 RNA.

Line 144. ‘subjected’ should be ‘subject’

Line 148. ‘in’ is missing between ‘and’ and ‘contrast’

Line 358. The word ‘membrane’ is missing.

Line 374. I don’t think Vm has been previously defined.

There are also many unnecessary uses of verbose or hyperbolic language eg ‘Astoundingly’

Author Response

Greetings dear reviewer 1, thank you for the guidance and assistance. Please find the cover letter at the attachment. Thank you very much dear reviewer 1. 

Reviewer 2 Report

The authors have presented a very good response to my questions and suggestions, and have made the appropriate changes to the manuscript. I think these have strengthened the manuscript. I do not have any additional comments or questions. 

Author Response

Thank you for the appreciation, guidance and assistance, dear reviewer 2.